# *Conformal Thinking*: Risk Control for Reasoning on a Compute Budget

**Xi Wang**[*][1]  **Anushri Suresh**[*][1]  **Alvin Zhang**[*][1]  **Rishi More**[*][1]  **William Jurayj**[1]  **Mehrdad Farajtabar**[2]
**Daniel Khashabi**[1]  **Eric Nalisnick**[1]

## Abstract

Reasoning Large Language Models (LLMs) enable test-time scaling, with dataset-level accuracy improving as the token budget increases, motivating *adaptive* reasoning—spending tokens when they improve reliability and stopping early when additional computation is unlikely to help. However, setting the token budget, as well as the threshold for adaptive reasoning, is a practical challenge that entails a fundamental risk-accuracy trade-off. We re-frame the budget setting problem as risk control, limiting the error rate while minimizing compute. Our framework introduces an *upper* threshold that stops reasoning when the model is confident (risking incorrect output) and a novel parametric *lower* threshold that preemptively stops unsolvable instances (risking premature stoppage). Given a target risk and a validation set, we use distribution-free risk control to optimally specify these stopping mechanisms. Empirical results across diverse reasoning tasks and models demonstrate the effectiveness of our risk control approach, demonstrating computational efficiency gains from the lower threshold and ensemble stopping mechanisms, all while adhering to the user-specified risk target.

## 1. Introduction

Reasoning LLMs enable test-time scaling: Spending more reasoning tokens often yields better performance (DeepSeek-AI et al., 2025; Snell et al., 2024). However, a practical challenge of reasoning LLMs lies in inducing *adaptive* reasoning behavior that adjusts to instance difficulty—deciding when additional thinking is still useful versus wasteful. Recent works propose adaptive thinking mechanisms (Wang et al., 2025a; Yang et al., 2025; Fu et al.,

2025) by monitoring the reasoning model's uncertainty in the answer (e.g. by measuring confidence or entropy), and when uncertainty falls below a pre-defined threshold, the reasoning is halted. Adaptive thinking enables instance-dependent token budgets, since the reasoning effort required to reach a confident threshold varies by problem. However, adaptive thinking does not alleviate the practical challenge of setting the reasoning budget; it only converts the problem of setting a token budget into setting a threshold. In fact, setting a threshold could be trickier than setting a token budget since the threshold value is often uninterpretable and could lie in arbitrary ranges depending on how the uncertainty metric (Figure. 1, left).

In this paper, we provide a principled framework for setting stopping rules for adaptive reasoning, inspired by prior work on adaptive compute in early-exit architectures (Schuster et al., 2022; Jazbec et al., 2024). In particular, we leverage a key implication of test-time scaling: *any early termination of reasoning introduces a risk of error*. Based on this observation, we reframe the problem of setting a reasoning budget as choosing an acceptable level of risk, an interpretable quantity that directly interfaces with downstream decision-making. We use distribution-free risk control (Bates et al., 2021; Angelopoulos et al., 2024; 2025) to automatically map a user-specified risk to the corresponding criteria for terminating the reasoning chain.

Specifically, we delineate two complementary types of risks and their corresponding controlling mechanisms (illustrated in Fig. 2): false positive risk of thinking the model has the correct answer (controlled by an upper threshold of confidence), and false negative risk of thinking the current (and future) answers will be incorrect. This latter risk is controlled by the lower threshold, a novel mechanism that forces reasoning to respect a schedule of progress. These two types of risks and thresholds are related to different sources of inefficiency: the upper threshold reduces wasted tokens after the model has effectively converged (Eq. (10)), and the lower threshold avoids spending tokens when further reasoning is unlikely to help (Eq. (11)).

Our contributions are: (i) We introduce a collection of loss functions that capture different sources of inefficiency and errors from early stopping in reasoning LLMs (Sec. 4.2); (ii)

---
[*]Equal contribution  [1]Johns Hopkins University, Baltimore, Maryland, USA  [2]Apple, USA. Correspondence to: Xi Wang <xwang457@jh.edu>, Eric Nalisnick <nalisnick@jhu.edu>.

*Proceedings of the 43rd International Conference on Machine Learning*, Seoul, South Korea. PMLR 306, 2026. Copyright 2026 by the author(s).

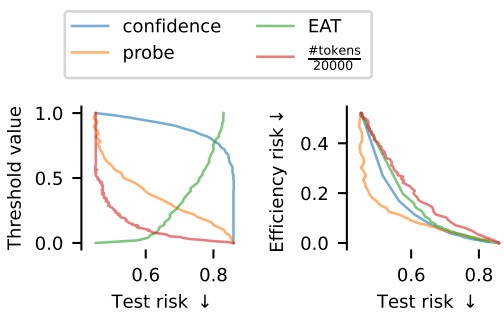

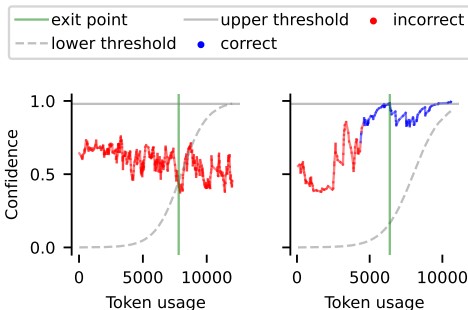

*Figure 1.* **Early-stopping behavior under different target test risks.** *Left:* Threshold values required to achieve a given target test risk vary substantially across early-stopping signals, indicating that threshold selection is signal-dependent. *Right:* The relative efficiency of different early-stopping methods depends on the target test risk, and no single method is uniformly most efficient across all risk levels. Results shown here use an upper-threshold-only stopping rule; introducing both upper and lower thresholds would further complicate the risk-threshold mapping.

We propose a novel parametric lower threshold that halts the reasoning when the model's confidence is increasing at a sufficient rate (Sec. 4.3); (iii) We demonstrate that, at the same target risk level, different approaches exhibit different efficiency, whereas our simultaneous upper and lower thresholds consistently yield efficiency gains (Sec. 5.3).

## 2. Background: Adaptive Early Stopping via Confidence (Upper) Threshold

**Reasoning models overthink.** Recent reasoning LLMs are post-trained to output an explicit reasoning trace in a delimited format, followed by a final answer:

$$y = \langle\texttt{think}\rangle \, r_{1:T} \, \langle\texttt{/think}\rangle \, a, \qquad (1)$$

where a complete generation $y$ is composed of: $r_{1:T}$ the reasoning segment of length $T$ (tokens or steps), special beginning/end of thinking tokens <think> and </think>, and $a$ the final answer. A common observation is that $T$ can be much larger than is necessary for many instances ("overthinking"), in turn introducing unnecessary inference cost. Models often continue to reason when a correct answer can already be elicited (Wang et al., 2025a; Yang et al., 2025), making $T$ a crucial hyperparameter.

**Uncertainty monitoring for adaptive early stopping.** Given an input $x$, let $r_{1:t}$ denote the partial reasoning trajectory upto $t$ steps. Recent works (Wang et al., 2025a; Yang et al., 2025) propose to adaptively early stop the reasoning by monitoring a scalar confidence or uncertainty signal computed from the partial trajectory:

$$s_t = u(x, r_{1:t}), \qquad (2)$$

where a large value of $s_t$ typically indicates higher confidence (or lower uncertainty, depending on convention).

*Figure 2.* **Dual-threshold early exit via risk-controlled confidence dynamics.** We plot confidence trajectories as a function of token usage under Qwen3-8B on AIME questions. **Left:** an unsolvable instance, model confidence fluctuates and fails to reach the upper threshold; the reasoning is halted early by the *parametric lower threshold*, preventing unnecessary token consumption. **Right:** a solvable instance, where confidence steadily increases and crosses the *upper threshold*, triggering termination once sufficient confidence is achieved.

Common choices for $s_t$ are derived from the statistical properties of tokens generated after $\langle\texttt{/think}\rangle$, such as their entropy (Wang et al., 2025a) or confidence (Yang et al., 2025). Per-time-point evaluations of $s_t$ can be noisy or have inconvenient ranges, and in turn, transformations such as smoothing, reciprocal, or normalization are often applied to reduce variance:

$$\tilde{s}_t = g(s_{1:t}), \qquad (3)$$

where $g$ denotes one of the aforementioned transformations that improves the (application-dependent) signal quality. Using this stabilized signal, the canonical early-exit policy halts reasoning at the earliest time the transformed score *exceeds* a threshold $\lambda$:

$$\tau = \min\Big\{t \geq 1 \,:\, \tilde{s}_t \geq \lambda\Big\}. \qquad (4)$$

We refer to this as the *upper threshold exit* mechanism. The policy then emits the answer $a$ and avoids generating $r_{t>\text{exit}}$. This framing unifies a broad set of "stop-when-confident" approaches (Wang et al., 2025a; Yang et al., 2025). Note that related "stop-when-stable" approaches (Liu & Wang, 2025; Wu et al., 2025)—which stop when the answer is stable across reasoning steps—do not disambiguate cases that can be stopped because the answer is correct vs the problem is hopelessly difficult, which is one of our core motivations.

**Advantages of adaptive thinking.** Given a dataset, adaptive reasoning assigns a different number of tokens to each question depending on how long it takes for the confidence to reach the target threshold. This allows better allocation of the budget across instances, in contrast to assigning a fixed budget to all questions, and as a result, adaptive thinking achieves the same dataset level accuracy with fewer total tokens, i.e., more efficient reasoning.

## 3. Related Works

**Early Exiting from Reasoning.** Much work has been done to reduce inference cost by *early exiting* chain-of-thought once additional reasoning is unlikely to help. As described in Section 2, most existing methods instantiate an *upper-threshold* rule: they monitor a confidence/uncertainty proxy (e.g., entropy or stability of intermediate answers) and stop when the model appears sufficiently certain or converged, via entropy signals, trial-answer confidence, or answer-consistency/convergence heuristics (Wang et al., 2025a; Yang et al., 2025; Mao et al., 2025; Liu & Wang, 2025; Liao et al., 2025; Wei et al.; Fu et al., 2025; Jurayj et al., 2025). **How we differ:** Our framework departs from this prior work in two ways. Firstly, we formalize a *lower-confidence* stopping criterion such that we "stop when reasoning is not becoming (sufficiently) more confident". This lower threshold can be used independently or combined with the upper into a dual-threshold rule that captures confident success *and* confident failure within a *single* trajectory. In fact, Wang et al. (2025a) explicitly state the limitation that, on challenging datasets, most problems never reach the target upper threshold, resulting in a great deal of wasted tokens. Secondly, prior work typically relies on hand-tuned cutoffs, sweeps, or heuristic criteria. In contrast, we use distribution-free risk control (Bates et al., 2021; Angelopoulos et al., 2024; 2025) to select thresholds that control target metrics with finite-sample guarantees.

**Risk Control for Reasoning LLMs.** Yet we are not the only work to combine risk control and LLM reasoning. The closest prior work is *Thought Calibration* (Wu et al., 2025), which is another "stop-when-stable" approach as it uses linear probes to predict when the answer at time $t$ is the same as the ultimate answer at time $T$. The authors then use learn-then-test (Angelopoulos et al., 2025) to calibrate the threshold at which the probe's output results in early termination of the reasoning chain. Note that Wu et al. (2025) also found that their approach capable of terminating unsolvable instances, similiar to our lower threshold mechanism; however this is based on their consistency-based probing signal, whereas our approach targets generic signals. *PAC Reasoning* (Zeng et al., 2026) uses risk control to decide when to use the output of a reasoning chain vs to forgo any reasoning. **How we differ:** *Thought Calibration* does not explicitly consider stopping when the problem is too difficult, only when the intermediate solution is either correct or stable (i.e. single threshold). *PAC Reasoning* does not control the length of the reasoning chain itself, only the decision to reason or not.

**Overthinking in reasoning.** Our motivation shares similarity with recent evidence that longer reasoning is not always beneficial, which shows that shorter completed chains can sometimes be more accurate or more efficient, either through multi-sample selection (Hassid et al., 2026), structured decoding/search (Xu et al., 2026), or broader reasoning-system pipelines (Moshkov et al., 2025). We view these findings as complementary motivation for adaptive stopping. Unlike these approaches, our method does not select among completed trajectories or redesign decoding; instead, it provides an instance-level stopping rule for an ongoing reasoning trajectory, with thresholds calibrated to meet an explicit risk target under a compute budget.

**Risk Control for Early-Exit Neural Networks.** Deciding when to stop a reasoning chain is conceptually identical to the older problem of deciding at which exit to stop an early-exit neural network (EENN) (Huang et al., 2018). Risk control has been applied to EENNs—for language modeling (Schuster et al., 2022), for time series classification (Ringel et al., 2024), and for image generation, semantic segmentation, and speculative decoding (Jazbec et al., 2024). Our work is primarily inspired by Jazbec et al. (2024)'s formalism, with the aim of applying it to LLM reasoning.

## 4. Method: Conformal Thinking

**Notation.** Let $y^*$ denote the ground true answer, $f_t(x)$ be the model's prediction at step $t$ given question $x$, and $T$ be the total reasoning steps. At each step $t$, the model also emits an auxiliary scalar signal $s_t \in \mathbb{R}$ (e.g., confidence), which is the primary input into an early stopping policy.

**Motivation and Goals.** Our work is motivated by two primary goals: (i) develop an adaptive early-stopping mechanism for LLM reasoning that allows user-specified control of the error rate(s), and (ii) extend existing upper-threshold-only approaches (Jazbec et al., 2024) to also stop when an instance is likely to be unsolvable within $T$ reasoning steps (our maximum budget). Regarding (i), for some loss function $\ell(y^*, f_t(x))$ that measures the quality of the intermediate solution $f_t(x)$, we ideally wish to control the expected loss, a.k.a. true risk (not empirical), as a function of the exiting hyperparameter $\lambda$:

$$\mathcal{R}(\lambda) \triangleq \mathbb{E}_{x,y^*} \left[ \ell(y^*, f_t(x); \lambda) \right] \leq \epsilon,$$

where $\epsilon \in (0,1)$ and $\lambda \in (0,1)$. The $\epsilon$ parameter, i.e., risk tolerance, is specified by the user—for example, as we shall consider, the rate at which early-terminated solutions are wrong (false positives). As $\lambda$ will be selected from a finite calibration set, controlling the true risk can only be done probabilistically, as we will describe in Section 4.4. As mentioned in Section 2, one way to specify $\lambda$ is via a confidence threshold. Yet this brings us to motivation (ii): upper-threshold approaches aim to stop when the reasoning chain has arrived at a correct solution. Given the challenging tasks to which LLMs are applied (e.g. *Humanity's Last Exam* (Center for AI Safety et al., 2026), cutting-edge mathematics), *we also want the reasoning to terminate when the*

*task is too hard*, meaning the LLM is unlikely to arrive at the correct answer before $T$ steps of reasoning.

## 4.1. Two-Threshold Early Exiting with Risk Control

Given our goals outlined above, we modify the existing one-threshold policy in Equation 4 to instead use two thresholds. Let $\lambda_+ \in (0,1)$ denote the *upper threshold*, which, as in previous work, aims to capture when the model is sufficiently confident it has arrived at the correct answer. Let $\lambda_- \in (0,1)$ denote the *lower threshold*, which encodes the minimum confidence level needed in order to proceed with reasoning. Our early-exiting policy can then be written as:

$$\tau = \min\left\{t \geq 1 : \tilde{s}_t \geq \lambda_+ \vee \tilde{s}_t \leq \lambda_-\right\} \quad (5)$$

where the thresholds will always be ordered such that $\lambda_+ > \lambda_-$. This ordering implies that the 'or' condition is exclusive; the model can only exit via one threshold.

We then wish to set $\lambda_+$ and $\lambda_-$ such that the risk,

$$\begin{aligned}\mathcal{R}(\lambda_+, \lambda_-) &\triangleq \\ \mathbb{E}_{x,y^*} &\left[\ell^+\left(y^*, f_t(x); \lambda_+\right) + \ell^-\left(y^*, f_t(x); \lambda_-\right)\right]\end{aligned} \quad (6)$$

is again upper-bounded by a user-specified tolerance, $\mathcal{R}(\lambda_+, \lambda_-) \leq \epsilon$ where again $\epsilon \in (0,1)$. Alternatively, if the user has risk targets in mind for each type of exiting (lower vs upper), they could be controlled individually:

$$\begin{aligned}\mathcal{R}^+(\lambda_+, \lambda_-) &\triangleq \mathbb{E}_{x,y^*}\left[\ell^+\left(y^*, f_t(x); \lambda_+\right) | \lambda_-\right] \leq \epsilon^+ \\ \mathcal{R}^-(\lambda_+, \lambda_-) &\triangleq \mathbb{E}_{x,y^*}\left[\ell^-\left(y^*, f_t(x); \lambda_-\right) | \lambda_+\right] \leq \epsilon^-.\end{aligned} \quad (7)$$

Note that the upper and lower control problems are not independent since the samples that exit via one threshold affect the distribution of instances that could exit via the other threshold. Below we describe the details of our method, which we call *Conformal Thinking*, as it will use conformal risk control algorithms (Bates et al., 2021; Angelopoulos et al., 2024; 2025) to set $\lambda_+$ and $\lambda_-$ to achieve efficient LLM 'thinking', i.e. reasoning.

## 4.2. Losses for Early-Exit Reasoning

Below we define four loss functions that capture the correctness-efficiency tradeoff induced by early exiting. All losses have the range $[0, 1]$, as we consider only supervised classification tasks.

**Losses for Predictive Performance.** We first quantify the classification error incurred by premature stopping. Starting with an upper-threshold quantity, the risk of incurring **false positives** can be captured by the loss:

$$\begin{aligned}\ell^+(y^*, f_t(x), \tilde{s}_t(x); \lambda_+) = \\ \mathbb{I}\left[\tilde{s}_t(x) \geq \lambda_+\right] \cdot \mathbb{I}\left[f_t(x) \neq y^*\right],\end{aligned} \quad (8)$$

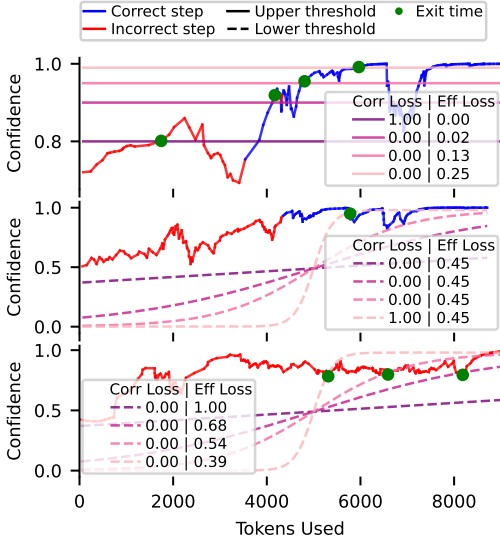

*Figure 3.* **Visualization of the proposed correctness and efficiency losses under different thresholds.** Lines of different colors (purple to pink) denote different threshold curves. Numbers in the box show the correctness and efficiency loss for each threshold. The top row shows the upper-threshold correctness and efficiency loss (Eq. (8) and (10)). Bottom two figures show lower-threshold sigmoid curves (Eq. (12)) and the corresponding losses the lower-threshold correctness loss (Eq.(9) and (11)). For a given instance, when multiple thresholds have the same correctness loss (e.g. top 3 lines in first row, all lines in the bottom row), the one with smallest efficiency loss is preferred, in order to maximize efficiency while maintaining user provided risk tolerance.

where $\mathbb{I}[\cdot]$ denotes the indicator function. The $\mathbb{I}\left[\tilde{s}_t(x) \geq \lambda_+\right]$ term checks if our confidence measure has exceeded the upper threshold, and $\mathbb{I}\left[f_t(x) \neq y^*\right]$ then checks if the current answer is correct.

For the lower-threshold loss, it should capture **false negatives**: $\tilde{s}_t(x)$ suggests reasoning will never arrive at the correct response in $T$ steps, but actually reasoning would return the correct answer at some step $k \in (t, T]$.

$$\begin{aligned}\ell^-(y^*, f_{t:T}(x), \tilde{s}_t(x); \lambda_-) = \\ \frac{\mathbb{I}[\tilde{s}_t(x) \leq \lambda_-]}{T - t + 1} \cdot \sum_{t \leq k \leq T} \mathbb{I}\left[f_k(x) = y^*\right].\end{aligned} \quad (9)$$

Notice this loss, unlike the one for false positives above, is farsighted. It not only needs to check if the current solution $f_t(x)$ is correct but all future solutions ($f_k(x)$, $t \leq k \leq T$) as well. The loss returns the accuracy as computed over the present and future solutions. Thus if there is only one future time point for which $f_k(x)$ is correct, this results in lower loss than if many future answers are correct.

**Losses for Efficiency.** Now we define losses that measure the efficiency of early stopping at step $t$. These losses will not depend on the threshold parameter $\lambda$ but rather be used to select a single $\lambda$ value when multiple would satisfy the

risk tolerance. Importantly, these losses are normalized w.r.t. the total reasoning budget $T$, since we want to quantify what fraction of $T$ was 'wasted' reasoning steps.

Again starting with the upper-threshold, we want to quantify the gap between the current and oracle stopping times, i.e. the index of the first reasoning step at which the answer is correct:

$$\mathcal{J}^+(t) = \frac{1}{T} \max(0, t - t'), \qquad (10)$$

where $t' = \min\{t \leq T : f_t(x) = y^*\}$. When a model arrives at a correct answer at step $t'$, any additional reasoning is wasted computation. Hence this loss measures normalized regret: the fraction of total steps spent deliberating after the answer was already correct. This loss complements the loss in Equation 8 since that loss only considers whether the answer is correct at the current exit and does not consider the counterfactual outcome of an earlier exit. Fig. 3, top row, shows both predictive and efficiency losses for several combinations of exits and upper threshold settings.

Moving on to the lower-threshold, this loss measures *what fraction of the previous exits had incorrect solutions?*. This loss complements the predictive loss in Equation 9 because it considers past exits:

$$\mathcal{J}^-(t) = \frac{1}{T} \sum_{k \leq t} \mathbb{I}[y^* \neq f_k(x)]. \qquad (11)$$

A low value suggests that either we stopped very early or the model solved the problem. A high value, (in the extreme case, 1), suggests that we used a large fraction of the budget without making much progress.

### 4.3. Dynamic Lower Threshold

We have so far described the lower threshold as, like the upper one, a static quantity. Yet notice that a static lower threshold can trigger the model to exit only when reasoning *decreases* its confidence. Rather, we wish to enforce the model to increase its confidence 'on a schedule'. If reasoning is not improving $\tilde{s}_t(x)$ at a particular rate, then we take this as a sign that the model likely will not arrive at a correct answer and we should exit to save tokens. Or in other words, the model must demonstrate a regular increase in confidence in order to be allowed to continue reasoning.

We enforce this confidence schedule by making the lower threshold a parametric function of the reasoning steps, meaning the lower threshold is *dynamic* rather than static. While many parameterizations are possible (and could be task dependent), we found general success with the following formulation. Given a total reasoning budget of $B$ tokens, let

---

**Algorithm 1** Computing hyperparameter(s)

---
**Require:** Validation set $\mathcal{V} = \{(x_n, y_n)\}_{n=1}^N$
**Require:** Risk budget $\epsilon \in (0, 1)$
**Require:** Candidate uncertainty signal set $\mathcal{S}$
**Require:** Threshold grids $\{\Lambda_s\}_{s \in \mathcal{S}}$
**Require:** Prediction procedure that yields a model output $\hat{y}_n$ for each $x_n$
**Require:** Risk estimator $\widetilde{\mathrm{Risk}}(\lambda; \mathcal{V})$
**Require:** Efficiency loss estimator $\widehat{\mathcal{J}}(s, \lambda; \mathcal{V})$
**Ensure:** Selected signal–threshold pair $(s^\star, \lambda^\star)$
1: **Initialize feasible set:** $\mathfrak{C} \leftarrow \emptyset$
2: **for** each signal $s \in \mathcal{S}$ **do** $\qquad \triangleright$*Grid search over signals*
3:      **for** each threshold $\lambda \in \Lambda_s$ **do** $\qquad \triangleright$*and thresholds*
4:          Compute adjusted estimated risk: $r \leftarrow \widetilde{\mathrm{Risk}}(\lambda; \mathcal{V})$
5:          **if** $r \leq \epsilon$ **then**
6:              Compute efficiency loss: $e \leftarrow \widehat{\mathcal{J}}(s, \lambda; \mathcal{V})$
7:              Add $(s, \lambda, e)$ to $\mathfrak{C}$
8: **if** $\mathfrak{C} = \emptyset$ **then** $\qquad\qquad\qquad \triangleright$*No feasible pair*
9:      **return** (NONE, NONE)
10: **else**
11:      $(s^\star, \lambda^\star) \leftarrow \arg\min_{(s,\lambda,e) \in \mathfrak{C}} e$ $\qquad \triangleright$*Select optimal pair.*
12:      **return** $(s^\star, \lambda^\star)$

---

$\omega_t$ be the total tokens generated up to reasoning step $t$.

$$\lambda_-(t; c, s, l, u) = \sigma(c(\omega_t - sB), l, u), \qquad (12)$$

$$\sigma(z, l, u) = \frac{u - l}{1 + e^{-z}} + l \qquad (13)$$

where $c \in \mathbb{R}^+$ denotes the slope, $\sigma(z, l, u)$ is the logistic (sigmoid) function squeezed into $(l, u)$, $l < u < 1$ (where $u$ can be typically chosen as $\lambda_+$), and $s \in \mathbb{R}$ shift the sigmoid horizontally relative to max token budget. Larger values of $c$ correspond to the need for faster increases in confidence to continue reasoning; combined with different values of $s$, Eq. (13) recovers a variety of shapes, e.g. linear ($s = 0.5, cB \ll 1$), exponential ($s > 1$), log ($s < 0$) or constant ($c \to 0$).

Now $\{c, s, l\}$ is the parameter we actually need to set, making our two-threshold risk in Equation 6 better written as $\mathcal{R}(\lambda_+, \{c, s, l\})$. See Fig. 3, bottom two rows, for various combinations of exits and dynamic threshold settings and their corresponding loss values under $s = 0.5$, $l = 0$, and various values of $c$. Lastly, for the simplicity of notation, we will abuse notation and denote the parameter set $\{c, s, l\}$ with just one symbol $c$ for the rest of the paper.

### 4.4. Risk-Controlling Hyperparameter Selection

Now that we have defined the necessary building blocks (two-threshold exiting policy, loss functions, dynamic lower threshold), we are ready to describe how to select $c$ and $\lambda_+$ such that risk is controlled at level $\epsilon$. We leverage existing algorithms for distribution-free risk control (Bates et al., 2021; Angelopoulos et al., 2024; 2025). In particular, we turn to the *upper confidence bound* (UCB) approach of

Bates et al. (2021), which has been shown to be effective for risk-controlled EENNs (Jazbec et al., 2024).

**Control with High Probability.** Given a held-out data set $\mathcal{V} = \{(x_i, y_i^*)\}_{n=1}^N$, the UCB algorithm can return settings of the hyperparameters such that the (true) risk is controlled with high probability:

$$\mathbb{P}_{\mathcal{V}}\left( \mathcal{R}\left(\hat{\lambda}_+, \hat{c}\right) \leq \epsilon \right) \geq 1 - \delta \qquad (14)$$

for $\delta \in [0, 1]$. The randomness in $\mathbb{P}_{\mathcal{V}}$ is driven by the size of the calibration set $\mathcal{V}$, since we need to use it to choose $\hat{\lambda}_+$ and $\hat{c}$. The larger the calibration set, the better the empirical risk will approximate the true risk, in turn making the hyperparameter selection more robust. A technical condition of applying the UCB algorithm is that either the loss functions (Bates et al., 2021) or risks (Jazbec et al., 2024) must be monotonic as a function of the hyperparameter(s). We assume that our risks are monotonic in $c$ and $\lambda_+$.

**Two-Step Hyperparameter Selection.** To use lower and upper threshold jointly, while we could run risk control for all possible pairs of $c$ and $\lambda_+$ (e.g. using learn-then-test (Angelopoulos et al., 2025)), we found the following two-step, decoupled procedure to work just as well in practice, though with some loss of theoretical rigor. We first use UCB to select $\lambda_+$ at risk target $\epsilon^+$, and then assuming the upper-threshold is fixed, we select $c$ based on controlling the risk $\mathbb{E}\left[\ell^- | \hat{\lambda}_+\right] \leq \epsilon^-$. The selection of $c$ is still guaranteed to control risk since it is done after $\hat{\lambda}_+$ is already selected. The upper risk bound could be violated when the lower threshold early exits more solvable instances than unsolvable ones (i.e. the lower threshold mechanism is completely broken), as we derived in App. C. While this could happen in theory, we believe in practice this could only occur under very extreme distribution shift such that the $c$ found on the validation set breaks completely in test environment.

**High-Level Overview of Selection Algorithm.** We refer the reader to Bates et al. (2021) and our Appendix B for the full details of the UCB algorithm. Yet we briefly sketch the idea here. We do so at the the level of intuition since most risk control algorithms take the following general form, and users may wish to swap UCB for an alternative. For each confidence signal $s \in \mathcal{S}$ and for each threshold $\lambda \in \Lambda_s$ (or value of $c \in \mathcal{C}$), we compute the empirical risk under a given loss function:

$$\widehat{\mathrm{Risk}}(\lambda; \mathcal{V}) = \frac{1}{N} \sum_{n=1}^N \ell\left(y_n^*, f_\tau(x_n), s_\tau(x_n); \lambda\right), \quad (15)$$

where $\tau$ denotes the exit position induced by signal $s$ under hyperparameter $\lambda$. A native implementation—essentially standard cross-validation—would select all values of $\lambda$ such that $\widehat{\mathrm{Risk}}(\lambda; \mathcal{V}) \leq \epsilon$. However, since $\mathcal{V}$ is finite (and often

small), the empirical risk could be an optimistic estimate of the true risk; as a result, thresholds selected by naive cross-validation can *overfit* to the validation set such that the true risk is *not* controlled. The first column in Figure 4 illustrates this phenomenon: the naive calibration frequently yields realized test risk above the target line $y = x$. UCB (along with other algorithms for risk control) replace the empirical risk with an *adjusted* risk,

$$\widetilde{\mathrm{Risk}}(\lambda; \mathcal{V}) = \widehat{\mathrm{Risk}}(\lambda; \mathcal{V}) + \text{(finite-sample correction)},$$

where (finite-sample correction) is a function of the validation set size. Hyperparameter selection is then done by finding the values that result in adjusted risks that respect the bound, i.e. $\widetilde{\mathrm{Risk}}(\mathcal{V}, s, \lambda) \leq \epsilon$. The second column in Figure. 4 illustrates the effect of having the finite sample correction, where the test risk now consistently stays below the user-specified risk tolerance across resamplings of $\mathcal{V}$.

In case multiple feasible pairs of $(s, \lambda)$ exist (for example, different types of uncertainty signal and thresholds), we select the one that minimizes one of the aforementioned efficiency losses,

$$\widehat{\mathcal{J}}(s, \lambda; \mathcal{V}) = \frac{1}{N} \sum_{n=1}^N \mathcal{J}\left(\tau_n(s, \lambda)\right), \qquad (16)$$

where $\tau_n(s, \lambda)$ denotes the exit position for the $n$th validation sample under signal $s$ and threshold parameter $\lambda$. This routine picks the most efficient hyperparameter configuration that does not exceed the risk target.

**Putting it all together.** Algorithm 1 summarizes the full implementation. Our final selection rule is: (i) enumerate signal - threshold candidates on $\mathcal{V}$, (ii) enforce the risk budget using risk control to account for finite-sample error, and (iii) among all feasible candidates, choose the one that minimizes the efficiency loss estimate. This yields a simple, plug-in calibration interface for arbitrary signals while providing stronger protection against validation overfitting than naive threshold tuning.

## 4.5. Advantages of the Risk-Control Perspective

Note that there are two settings where risk control is *not* strictly necessary.

- **Purely comparative evaluation.** If the goal is only to demonstrate that one early stopping signal dominates another in the accuracy–compute trade-off, then a threshold sweep is sufficient: one can enumerate thresholds and report the resulting Pareto frontier (or area-under-curve) on a labeled benchmark, without committing to any particular operating point.
- **Perfectly interpretable signals.** If the stopping signal is directly calibrated to correctness, e.g., a confidence

value $p$ truly means $\Pr(\text{correct}) = p$, then choosing a threshold is equivalent to choosing an error rate, and ad-hoc thresholding becomes less problematic.

**Why these conditions are atypical in practice.** The first scenario diverges from deployment settings in two ways. First, real systems typically require *one* concrete configuration (or a small set of configurations) that meets a user- or application-specified error tolerance, rather than reporting an entire sweep curve post hoc. The second condition also rarely holds for existing early-stopping signals: The mapping from a raw threshold to the achieved error is highly signal-, model-, and task-dependent (Fig. 1, left). Moreover, the *lower* threshold, which is often implemented as a parametric function of time/tokens, contains more hyperparameters that are even harder to interpret and transfer across tasks, making hand-tuning unreliable and amplifying the need for principled, finite-sample-aware selection.

**What risk control buys us.** Our framework therefore, asks the user to specify a *risk tolerance* $\epsilon$, an operational quantity with downstream meaning, and uses a held-out validation set with finite-sample correction to automatically select thresholds (and, when applicable, choose among multiple stopping criteria) so that the realized test risk respects the user-specified tolerance with high probability. This shifts the burden from tuning opaque thresholds to selecting an interpretable error tolerance.

## 5. Empirical validation

In this section, we provide empirical verification of the effectiveness of risk control. Then we demonstrate the efficiency gain from the risk control framework: From ensembling signals and from combining upper and lower thresholds.

### 5.1. Experiment setup

**Models and datasets** We evaluate our methods on Qwen3-8B, Qwen3-30B-A3B, DeepSeek-R1-Distill-Qwen-32B, as well as Qwen3-VL-8B. For datasets, we consider AIME (1983-2025, 1011 samples), a subset of DeepScaleR (Luo et al., 2025, 1189 samples) with AIME removed, GPQA-Diamond (Rein et al., 2024, 198 samples) and MathVision (Wang et al., 2024, 304 samples), a vision language reasoning dataset.

**Reasoning generation** We use a system prompt of "Please reason step by step, and put your final answer within \boxed{}.". For all models, we use the recommended decoding configurations from the model cards. The generation is conducted using vLLM for all experiments.

**Answer and uncertainty signal elicitation** During the generation of reasoning, we consider text between two consecutive \n\n delimiters as a chunk; all uncertainty signals are

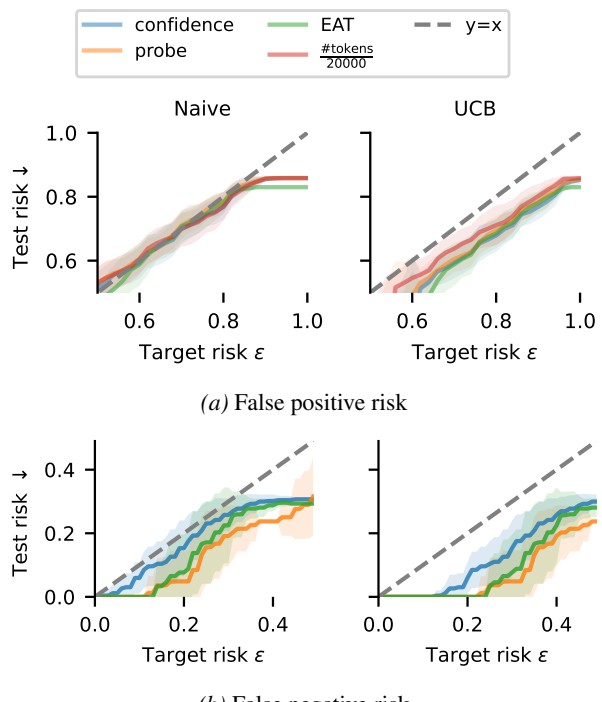

*(a)* False positive risk

*(b)* False negative risk

*Figure 4.* **Empirical verification of risk control.** We plot the empirical test risk (y-axis) against the user-specified target risk $\epsilon$ (x-axis). Solid lines and shaded regions indicate the mean and standard deviation over 40 random test-validation splits. Different colors denote different early-stopping signals. The left panel (NAIVE) selects thresholds on the validation set without finite-sample correction, leading to frequent violations in the realized test risk exceeding $\epsilon$, particularly on false negative risk (Eq. (9)) controlled by the lower threshold (Eq. (12)), which has more flexibility and therefore more prone to noise. The right panel (UCB) applies a probabilistic risk control procedure that accounts for validation uncertainty, guaranteeing that the test risk under the selected threshold is upper-bounded by $\epsilon$ with high probability.

computed at the end of each growing chunk. If we decide to early stop at a chunk, we append the forcing string `"\n **Final Answer**\n \boxed{"` to elicit a canonical boxed answer. We consider 2 uncertainty-based signals: Confidence (Yang et al., 2025) and EAT (Wang et al., 2025a); On Qwen3-8B, we trained a probe model (Zhang et al., 2025) on AIME. Additionally, we consider using the number of tokens as a stopping criterion. Appendix A.1 provides an introduction to the signals.

### 5.2. Risk control framework controls risk

We begin by performing a sanity check on the effectiveness of Alg. 1. In particular, we aim to verify, given a risk tolerance $\epsilon$, whether the threshold picked on the validation set gives a test risk smaller than $\epsilon$.

We consider Qwen3-8B model on AIME, under a variety of early stopping signals, including uncertainty-based, probe-

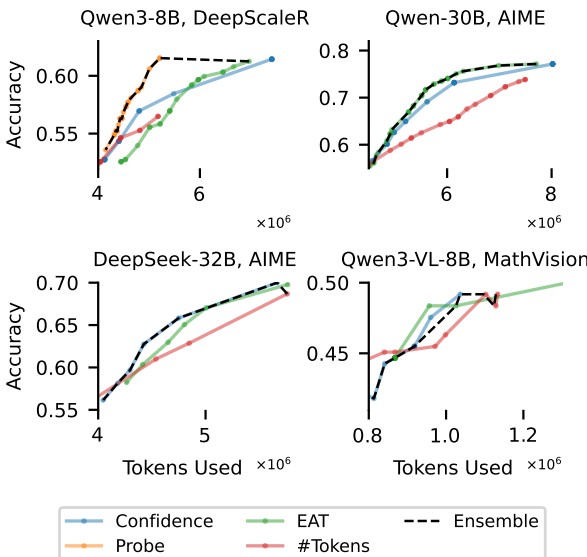

*Figure 5.* **Ensemble of signals improves efficiency.** Under four models, we consider upper-threshold only early stopping. Given a target tolerance $\epsilon$, risk control framework picks the signal that minimizes the efficiency loss (Eq. (10)), forming an *ensemble* of signals, which translates to superior efficiency on the test set (better accuracy v.s. token trade-off).

based, and just by the number of tokens. We randomly generate 40 validation-test splits, with a validation set size of 50 samples (5 percent), and we study both false positive (Eq. (8)) and false negative loss (Eq. (9)), We enumerate over values of $\epsilon$ between $[0, 1]$ with a step size of $0.01$, find the thresholds using Alg. 1, and then compare the achieved risk on the test set with $\epsilon$.

The results are demonstrated in Fig. 4. In particular, using finite sample correction (UCB) enables the risk on the unseen test set to lie below the user-specified threshold; Naive cross-validation, despite having the mean controlled, its standard deviation bands often cross the $y = x$ line, i.e., many individual runs exceed the target risk.

### 5.3. Risk control framework improves efficiency

Now we demonstrate how the risk control framework enables more *efficient* reasoning. The setting involves enumerating over epsilons, then for each epsilon, we check the total number of questions correctly answered by the model v.s. the total number of tokens used.

**Ensembling signal improves efficiency** Given an $\epsilon$, a critical step of Alg. 1 is to select a signal threshold pair that minimizes the efficiency loss (line 6 and 11). This enables us to construct an ensemble of uncertainty signals where we pick the most efficient signal and threshold for each $\epsilon$ and dataset. Our results confirm that the most efficient ones on

the validation set transfer to efficiency gain on the test set, as is shown in Fig. 5: Overall ensemble manages to picks the most efficient signal across all candidates, e.g. on Qwen3-8B, we have access to a powerful probing model trained on AIME, so ensemble consistently picks probe across $\epsilon$.

**Lower threshold improves efficiency.** Next we examine how the *upper* and *lower* thresholds complement each other in improving efficiency. The upper threshold mainly saves tokens on *solvable* instances by stopping once the model becomes confident, whereas the lower threshold mainly saves tokens on *unsolvable* instances by halting runs whose uncertainty fails to improve and would otherwise consume the full budget. Thus, the value of a lower threshold depends on the *solvable:unsolvable* composition of the test set. To study this dependence, we construct evaluation sets with controlled **solvable:unsolvable ratios** of $3:1$, $1:1$, and $1:3$. We pool AIME and GPQA, label each instance by whether the model can reach the correct final answer under the full token budget, and subsample to match the target ratio. Using confidence as the uncertainty signal, we compare UPPER-ONLY, LOWER-ONLY, and LOWER+UPPER. For UPPER-ONLY and LOWER-ONLY, each point on the curve corresponds to a risk tolerance $\epsilon$ calibrated on the validation split. For LOWER+UPPER, we fix the upper threshold to the value achieving the smallest validation risk and then sweep the lower-threshold.

As shown in Fig. 6, when solvable instances dominate (3:1), UPPER-ONLY captures most savings and LOWER-ONLY adds little. When unsolvable instances are common (1:1 and 1:3), UPPER-ONLY becomes inefficient: its curve stays near the high-token region because many runs never satisfy the confidence cutoff and therefore run to the maximum budget (Fig. 6, top row). Adding the lower threshold mitigates this failure mode and yields a clear leftward shift at comparable accuracy. The bottom row of Fig. 6 confirms this division of labor: at a representative operating point (second-highest accuracy for LOWER+UPPER), solvable instances mostly exit via the upper threshold, while unsolvable instances mostly exit via the lower threshold.

### 5.4. Ablation study

Lastly, we study the robustness of risk control under variation of validation set sizes as well as under distribution shift. We focus the study on Qwen3-8B evaluated on DeepScaleR.

**Size of validation set** We fix the test set size as $800$ samples and vary the validation set size in $\{8, 16, 40\}$. Broadly, we find that as validation set size decreases, the advantage of UCB over Naive becomes more pronounced (Fig. 7).

**Distribution shift** We study two types of distribution shift between the validation and test set:

- **Length shift**: The validation set has an overall length dif-

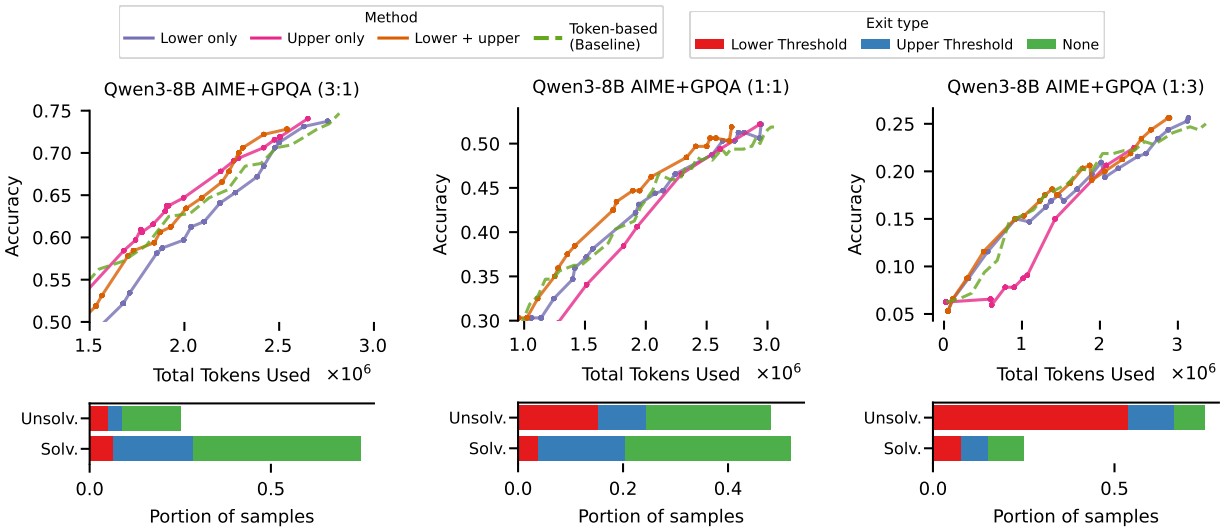

*Figure 6.* **Lower-threshold gains grow when unsolvable instances are prevalent.** We evaluate Qwen3-8B with confidence as the uncertainty signal on datasets with *solvable:unsolvable* ratios of 3:1, 1:1, and 1:3 (constructed by pooling AIME and GPQA, labeling instances by solvability under the full token budget, and subsampling to match each ratio). **Top:** test accuracy (instances abstained by lower threshold considered as wrong) versus total tokens for LOWER-ONLY, UPPER-ONLY, and LOWER+UPPER; each point corresponds to a validation-calibrated risk tolerance $\epsilon$, and we include a uniform token-budget baseline. When unsolvable instances are common (1:1 and 1:3), UPPER-ONLY clusters near the high-token regime because many runs never reach the confidence cutoff and thus run to the budget limit, while adding the lower threshold shifts the curve left (similar accuracy with fewer tokens). **Bottom:** for LOWER+UPPER at a representative operating point (second-highest accuracy), we report the solvable/unsolvable fractions and which condition triggered termination, showing that solvable instances typically exit via the upper threshold and unsolvable ones via the lower threshold.

ferent than the test set. Results are shown in Fig. 8: Short (validation) to long (test) brings challenges, particularly for the lower threshold risk, whose shape depends heavily on the reasoning length. The false positive risk controlled by the upper threshold does not show much degradation.

- **Dataset shift**: The validation set and the test set come from different datasets. We consider AIME v.s. GPQA-Diamond, i.e. Math v.s. Science questions. Note that this also includes difficulty shift, as GPQA has a much lower solved rate than AIME. Results are shown in Fig. 9. Again, without finite sample correction, Naive shows severe excess risk while UCB continues to bound the test risk below the target risk $\epsilon$.

# 6. Conclusions, Limitations, and Future Work

**Conclusion** We propose *Conformal Thinking*, a methodology to improve the efficiency of reasoning in LLMs. Instead of stopping the reasoning chain when it has reached high-confidence or stability, we add a second threshold that represents a 'schedule' by which the model must increase confidence in order to continue reasoning. This allows the reasoning chain to terminate in both cases of high confidence and cases for which it seems hopeless the model will be correct.

**Limitation** The primary limitation in our approach is that we assumed our risk functions are monotonic in the hyperpa-

rameters and that the risk from the upper loss is unaffected by the introduction of the lower threshold. Since both these assumptions are with respect to the true risk, they can be rigorously verified only in large-sample regimes (for which the finite-sample correction is less important). Additionally, our evaluation is currently limited to scientific and math reasoning tasks, mainly due to existing uncertainty signals not capable of early stopping tasks with less structured reasoning and output.

**Future work** Future work can consider instance-wise parameter / threshold setting using test time learning (Zhou et al., 2026); Applying risk control framework to agentic framework(e.g. (Sadhuka et al., 2025)) to recognize that an agent is going to fail; Extending the framework to alternative forms of test time scaling, e.g. parallel reasoning chains; Lastly, one could improve the calibration and interpretability of the uncertainty signal itself, e.g., by adding an auxiliary objective during post-training (Wang et al., 2026), thereby reducing the reliance on post-hoc calibration.

# Impact Statement

This paper presents work whose goal is to advance the field of Machine Learning. There are many potential societal consequences of our work, none which we feel must be specifically highlighted here.

## Acknowledgements

We thank Metod Jazbec and Alexander Timans for helpful feedback. This work is supposed in part by Apple and Defense Advance Research Projects Agency (DARPA) under Contract No. HR001125C0304 and ONR grant (N0001424-1-2089). Any opinions, findings and conclusions or recommendations expressed in this material are those of the author(s) and do not necessarily reflect the views of DARPA. We acknowledge the use of computational resources from the Johns Hopkins DSAI cluster.

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

# A. Extended experiment specifications.

## A.1. Signal Extraction

We evaluate confidence signals that measure uncertainty or confidence at each thought chunk. Specifically, we focus on two primary metrics: Entropy After </think> (Wang et al., 2025b, EAT) and Confidence (Yang et al., 2025).

Let $p_\theta(\cdot \mid x)$ denote the next-token distribution at prefix $x$. We define the transformed prefix $x_{\text{forced}}$ by appending both, the thought termination tag and a forcing string.

$$x_{\text{forced}} = x \oplus \text{</think>} \oplus \texttt{forcing\_string} \tag{17}$$

where `forcing_string`, as described in the main text, is `"\n **Final Answer**\n \boxed{"`.

For EAT, we additionally append the thought termination tag </think> before applying the forcing string.

**Confidence** greedily rolls out an answer after $x_{\text{forced}}$, denoted as $\mathbf{a} = (a_1, \ldots, a_L)$. Then the length-normalized log likelihood over $\mathbf{a}$ is used as the confidence score

$$C_{\text{seq}}(x, \mathbf{a}) = \frac{1}{L} \sum_{i=1}^{L} \log p_\theta(a_i \mid x, a_{<i}). \tag{18}$$

**EAT** does not generate a rollout, instead it directly looks at the entropy of the next token prediction distribution after $x_{\text{forced}}$:

$$\mathbb{H}(p_\theta(\cdot \mid x_{\text{forced}})). \tag{19}$$

where `forcing_string`, as described in the main text, is defined as `"\n **Final Answer**\n \boxed{"`

**Confidence** greedily rollouts an answer after $x_{\text{forced}}$, denoted as $\mathbf{a} = (a_1, \ldots, a_L)$. Then the length-normalized log likelihood over $\mathbf{a}$ is used as the confidence score

$$C_{\text{seq}}(x, \mathbf{a}) = \frac{1}{L} \sum_{i=1}^{L} \log p_\theta(a_i \mid x, a_{<i}). \tag{20}$$

**EAT** does not generate rollout, instead it directly looks at the entropy of the next token prediction distribution after $x_{\text{forced}}$:

$$\mathbb{H}(p_\theta(\cdot \mid x_{\text{forced}})). \tag{21}$$

**Probe signals.** We additionally compute probe-based signals trained on AIME reasoning trajectories.

Representation extraction and labeling: Let $P$ denote the original problem prompt and $x_{1:T}$ the trajectory. Suppose the model emits $K$ candidate final answers at token indices $1 \leq t_1 < \cdots < t_K \leq t_T$, with the $s$-th candidate answer denoted $\hat{a}_s$. For each step $s$, we reconstruct the decoding context

$$C_s = [P; x_{1:t_s}],$$

and extract the hidden representation of the last token at the last hidden layer:

$$h_s \in \mathbb{R}^d.$$

Each step is labeled according to correctness relative to the gold answer $a^\star$:

$$y_s = I[\hat{a}_s = a^\star] \in \{0, 1\}.$$

This yields a dataset

$$\mathcal{D} = \{(h_s, y_s)\}_{s=1}^{K}.$$

Probe model and training: We train a two-layer MLP probe (Zhang et al., 2025) to predict stepwise correctness $y_s$ from the representation $h_s$. The probe is trained on AIME 1983–2024.

## B. Risk control and finite-sample correction

This section details how we calibrate threshold parameters using distribution-free risk control. The goal is to select a signal–threshold pair (or parameter) that (i) satisfies a user-specified risk budget $\epsilon$, and (ii) among all feasible candidates, minimizes the corresponding efficiency loss.

### B.1. Problem setup

Let $\mathcal{V} = \{(x_i, y_i^*)\}_{i=1}^n$ be a labeled validation set. For a given uncertainty signal $s \in \mathcal{S}$ and a threshold parameter $\lambda \in \Lambda_s$ (e.g., $\lambda = \lambda_+$ for the upper threshold, or $\lambda = c$ for the parametric lower threshold), let $\tau_i(s, \lambda)$ denote the induced exit time on instance $i$. Given an instance-wise loss $\ell(\cdot; \lambda) \in [0, 1]$ (e.g., Eq. (8) or Eq. (9)), the population risk is

$$\text{Risk}(s, \lambda) = \mathbb{E}\big[\ell\big(y^*, f_{\tau(s,\lambda)}(x), \tilde{s}_{\tau(s,\lambda)}(x); \lambda\big)\big],$$

and its empirical estimate on $\mathcal{V}$ is

$$\widehat{\text{Risk}}(\mathcal{V}, s, \lambda) = \frac{1}{n} \sum_{i=1}^n \ell\big(y_i^*, f_{\tau_i(s,\lambda)}(x_i), \tilde{s}_{\tau_i(s,\lambda)}(x_i); \lambda\big). \tag{22}$$

### B.2. Why finite-sample correction is needed

Using $\widehat{\text{Risk}}(\mathcal{V}, s, \lambda)$ directly can be over-optimistic due to sampling noise: a candidate $(s, \lambda)$ that appears feasible on $\mathcal{V}$ may violate the target risk after deployment. This issue is amplified when scanning a large grid $\bigcup_{s \in \mathcal{S}} \Lambda_s$ and selecting the most efficient feasible configuration, since the selection procedure can exploit random downward fluctuations in the empirical risk.

To mitigate this, we replace the raw empirical estimate with a conservative, finite-sample-adjusted quantity $\widetilde{\text{Risk}}(\mathcal{V}, s, \lambda)$, and enforce feasibility using $\widetilde{\text{Risk}}(\mathcal{V}, s, \lambda) \leq \epsilon$.

### B.3. Calibration methods

We consider two variants that differ only in how the risk constraint is enforced.

**Naive (no finite-sample correction).**
$$\widetilde{\text{Risk}}_{\text{naive}}(\mathcal{V}, s, \lambda) = \widehat{\text{Risk}}(\mathcal{V}, s, \lambda). \tag{23}$$

This baseline is often effective when $n$ is large, but provides no protection against validation overfitting.

**UCB: concentration-based upper confidence bound.** For losses bounded in $[0, 1]$, Hoeffding's inequality implies that with probability at least $1 - \delta$,

$$\text{Risk}(s, \lambda) \leq \widehat{\text{Risk}}(\mathcal{V}, s, \lambda) + \sqrt{\frac{\log(1/\delta)}{2n}},$$

for a fixed $(s, \lambda)$. We therefore define

$$\widetilde{\text{Risk}}_{\text{UCB}}(\mathcal{V}, s, \lambda) = \widehat{\text{Risk}}(\mathcal{V}, s, \lambda) + \sqrt{\frac{\log(1/\delta)}{2n}}, \tag{24}$$

and declare $(s, \lambda)$ feasible only if $\widetilde{\text{Risk}}_{\text{UCB}}(\mathcal{V}, s, \lambda) \leq \epsilon$. Operationally, the correction is larger when $n$ is small and vanishes as $n$ grows, yielding an adaptive conservativeness.

**Practical remark (multiple comparisons).** Eq. (24) provides a high-probability bound for a *fixed* candidate. When scanning a finite grid $\mathcal{G} = \{(s, \lambda) : s \in \mathcal{S}, \lambda \in \Lambda_s\}$, one may strengthen the bound via a union bound by replacing $\delta$ with $\delta/|\mathcal{G}|$. In our experiments, we use the simple correction above and report realized risks on a held-out test set.

## C. Violation of false positive guarantee

The false positive rate is defined as

$$\text{FPR} = \frac{I}{I + C}, \tag{25}$$

where $I$ and $C$ denote the numbers of incorrectly and correctly answered queries, respectively.

If the lower threshold causes us to abstain from fractions $p_i$ and $p_c$ of incorrect and correct samples, respectively, then the new false positive rate becomes

$$\text{FPR}_{\text{new}} = \frac{(1 - p_i)I}{(1 - p_i)I + (1 - p_c)C}. \tag{26}$$

It follows that when $p_i < p_c$, i.e., the lower threshold filters out proportionally more correct samples than incorrect ones, we have

$$\text{FPR}_{\text{new}} > \text{FPR}, \tag{27}$$

meaning that the guarantee breaks. Conversely, when $p_i \geq p_c$, the guarantee remains valid.

This failure mode arises when the lower-threshold mechanism itself fails, for example due to a significant distribution shift between the calibration and test sets. In such cases, the lower threshold may have been well calibrated on the calibration set but generalizes poorly to the test set, causing it to disproportionately abstain on correct samples. It is worth noting, however, that distribution shift tends to break all guarantees, not just this one. We have added an explicit limitations section discussing this issue; see also our response to Reviewer RnRS.

In practice, as long as the lower threshold functions as intended, filtering out more incorrect samples than correct ones, i.e.,

$$p_i \geq p_c, \tag{28}$$

the upper-threshold guarantee remains intact.

## D. Ablation study results

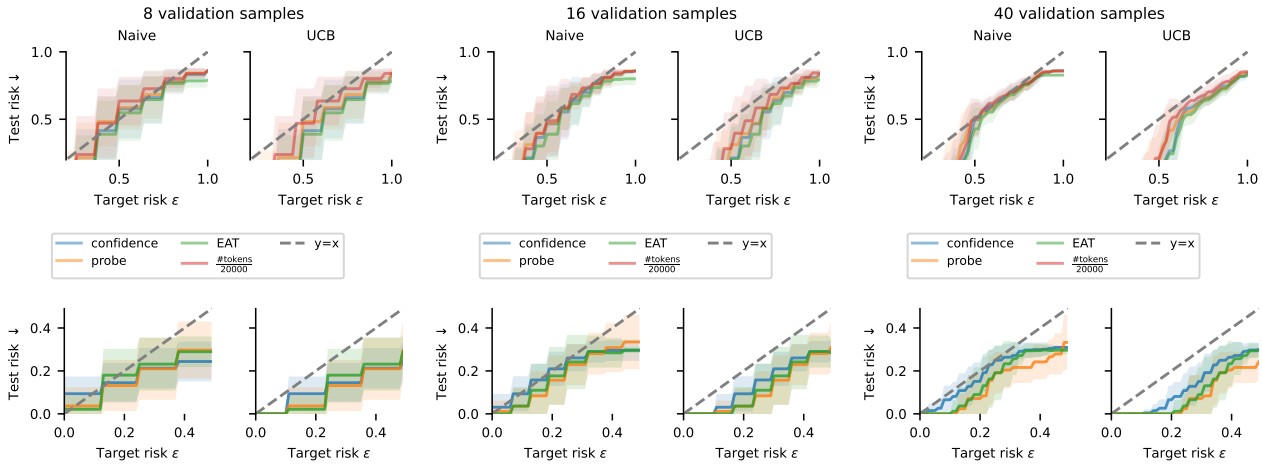

*Figure 7.* Ablation on validation set size. **Top row:** False positive risk; **Bottom row:** False negative risk. Principled risk control approach (UCB) shows better risk control than Naive cross-validation under small validation set size, as well as for

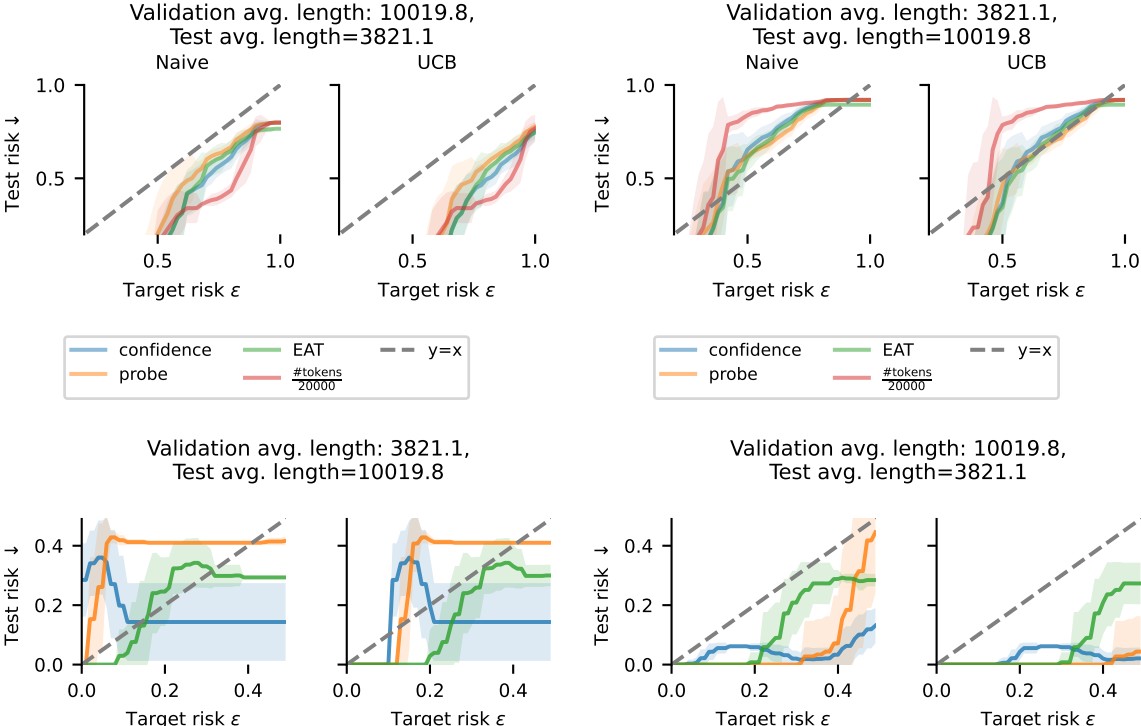

*Figure 8.* Ablation on length shift between validation and test set. **Top row:** False positive risk; **Bottom row:** False negative risk. Short to long shift (first column) brings more challenges to risk control. For upper threshold (top row), principled risk control alleviates excessive risk for all signals except for token-based. False negative risk, controlled by the lower threshold, shows a lack of robustness against length shift, in that the shape of the lower threshold is dependent on the horizon of the reaosning.

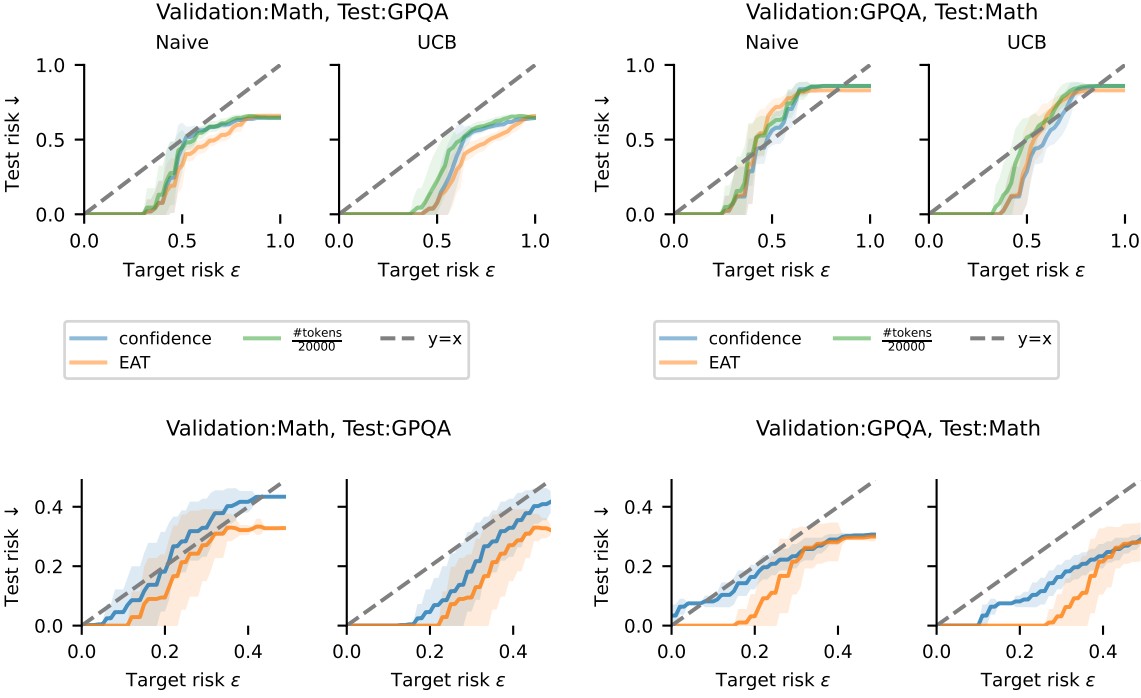

*Figure 9.* Ablation on dataset shift between validation and test set. **Top row:** False positive risk; **Bottom row:** False negative risk. Consistent with previous observations, using principled risk control again yields more controlled risk under distribution shift.

