# OpenReview forum: "Conformal Thinking: Risk Control for Reasoning on a Compute Budget"
_ICML.cc/2026/Conference — ICML 2026 regular_

### Official Review · Reviewer_EgaZ · 2026-03-10

**Soundness:** 2
**Presentation:** 3
**Significance:** 3
**Originality:** 2
**Overall Recommendation:** 4
**Confidence:** 3

**Summary:**

This paper reframes early stopping for reasoning LLMs as a risk control problem where the user specifies an interpretable risk tolerance and the framework automatically selects the stopping thresholds via distribution-free calibration on a labeled validation set. The framework includes two complementary stopping mechanisms: (1) an upper threshold that halts reasoning once the model appears confident, and (2) a parametric lower threshold that terminates reasoning when the model fails to make progress toward a solution. Both thresholds are calibrated with a finite-sample correction to prevent overfitting. Experiments across models and benchmarks show that finite-sample correction keeps test risk below the user-specified target, and that combining both the upper and lower threshold yields consistent efficiency gains, especially when many instances are unsolvable.

**Compliance With Llm Reviewing Policy:**

Affirmed.

**Key Questions For Authors:**

Q1. In dual-threshold setting, how frequently does the false-positive guarantee get violated in practice, and what is the typical magnitude of the violation?

Q2. Did you try alternative lower-threshold parameterizations (e.g., linear, piecewise-linear, exponential)? If not, what specifically motivates the sigmoid choice?

Q3. How sensitive are the results to the false-negative loss definition in Eq. (7)? For example, do alternatives such as those mentioned in W3 materially change the lower-threshold behavior?

**Limitations:**

Yes

**Strengths And Weaknesses:**

Strengths:

S1. The problem formulation is practically motivated. Replacing opaque threshold tuning with an interpretable risk tolerance better matches how users specify deployment requirements.

S2. Existing adaptive reasoning methods focus almost on the “stop when confident” pattern. This paper identifies and formalizes the complementary failure mode of exhausting the full reasoning budget on instances the model is unlikely to solve.

S3. The framework is signal-agnostic by design and can operate on any scalar stopping signals beyond the ones evaluated in the paper.

S4. Thresholds are calibrated in a distribution-free manner with finite-sample correction, which allows the method to work well even with a small validation set and makes it practical for deployment.

Weakness:

W1. The lower threshold is the main novelty of the paper, but its sigmoid parameterization seems ad hoc. The paper does not explain why this form is appropriate, why it should be centered at B/2, or how sensitive the method is to this choice.

W2. Prior work (Wu et al., 2025) already noted that consistency-based probes can implicitly prioritize terminating unsolvable instances. However, the paper does not clearly explain what additional benefit the explicit lower threshold provides beyond this implicit effect, nor does it include a direct empirical comparison.

W3. Choices of the false-negative loss (Eq. (7)) is not well justified. Intuitive alternatives such as eventual success under the full budget or discounted future progress could lead to different lower-threshold behavior. The paper should better motivate this definition or compare it against such alternatives.

W4. The paper notes that the false-positive guarantee may break when upper and lower thresholds are combined, but this is treated heuristically. Since the dual-threshold setting is the main use case, the paper should quantify how often and by how much the false-positive rate exceeds the target, and when joint calibration is worth given its significant cost.

W5. The losses and experiments are limited to exact-match tasks (e.g., math/science reasoning). It remains unclear how well the framework extends to broader settings such as coding or open-ended generation.

---

> ### Author Rebuttal · Authors · 2026-03-30
>
> We would like to thank the reviewers for the detailed and positive comments. We especially appreciate the reviewer for acknowledging our work’s practical value, novel lower threshold mechanism, signal-agnostic design, and the principled finite-sample correction when using a validation set.
>
> # Sigmoid shape (W1, Q2)
>
> First, we would like to apologize that there is a typo in Eq.12. In our experiments, we actually have three parameters for the sigmoid:
>
> - The slope: Controls the steepness of the transition — a larger slope produces a sharper, more abrupt switch, while a smaller slope yields a more gradual curve.
> - The shift: Controls the location of the inflection point along the token axis, i.e., at which token count the transition occurs.
> - The lower bound: Sets the minimum value of the function, allowing the curve to start from a non-zero baseline.
>
> Y = sigmoid(x) * (1 - lower_bound) + lower_bound, x = (#tokens - shift) * slope
>
> It is only for the visualization part (Fig. 3), we keep the lower bound as 0, shift as B / 2 and only adjust the slope.
>
> For the early stopping experiments,  we search over all three parameters. This added flexibility allows the resulting function to cover a very wide range of shapes, for example:
> - A shift far to the left vs. right of the token budget places the inflection point outside the operating range, causing the function to behave as a purely convex vs. concave curve over the token range of interest.
> - A small slope, large shift, and positive lower bound causes the sigmoid to flatten out within the token range, effectively recovering a constant lower threshold.
>
> We apologize again for the typo and have corrected it in the revised manuscript.
>
> # Comparisons with Wu et al., 2025  (W2)
>
> We believe the implicit termination of unsolvable instances requires training a specific type of probe that checks the partial reasoning chain’s answer consistency with the final answer, where high consistency indicates reasoning trapped in a loop.
>
> Our method targets generic uncertainty signals without such a property.
>
> # Design of false negative loss (W3)
>
>
> Our original motivation for the false negative loss is described from at line 195 - 207, left column. In particular, the false negative loss penalizes the model for halting on instances it would have solved correctly with more computation. Crucially, Eq. (7) measures correctness averaged over all future steps rather than penalizing based on the mere existence of a correct future step. This means a model that reaches the correct answer only sporadically — e.g. through random guessing — incurs little loss for halting early, since its future correctness is unreliable. Only when the model would answer correctly consistently does early halting incur a high penalty, ensuring the loss encourages continued reasoning only when doing so yields reliable gains.
>
> However, we agree that this is an important issue and comparisons with other losses are worth including. We have included additional discussions, comparisons with eventual success:
>
> Performance on early stopping (settings in Fig.6, middle column) experiments is shown at https://imgur.com/a/TNmk4dV
>
> The risk value (averaged loss over validation set) under different function values for the two losses is shown at https://imgur.com/a/KuG6CFF
>
> Broadly, we did not see a significant difference. We believe this is because the actual loss that picks the function parameter is an aggregation over all data points, and the extremely sparse eventual success we worried about may only happen rarely in practice, so the two losses differ little in the cases we considered.
>
>
> # Violation of false positive guarantee (W4 and Q1)
>
> Please refer to our response to reviewer mCrS
>
> https://openreview.net/forum?id=noDJPmA3ha&noteId=wx9Jz5Ns4A
>
> In short, the upper threshold guarantee would break if the lower threshold mechanisms is “completely broken”, i.e., rejecting more solvable than unsolvable ones. However, we do not expect this to happen often in practice unless under a very extreme distribution shift.
>
> # More tasks (W5)
>
> Firstly, estimating the uncertainty over free-form rollout is hard and we expect these lightweight uncertainty signals to work suboptimally on open-ended generation.
> With that said, when such signals are available, we will update the experiments and manuscript accordingly to test them out.
>
> However, unreliability of the signal does not pose a challenge to our framework: If the uncertainty signal does not work on a particular domain, our Alg 1 would fall back to the token-based stopping criteria, as no threshold for the uncertainty signal can meet the user-specified risk.

---

> > ### Author Rebuttal · Reviewer_EgaZ · 2026-04-04
> >
> > My concerns are all resolved.

---

> > > ### Author Response · Authors · 2026-04-06
> > >
> > > We would like to thank the reviewer again for the very thoughtful and detailed feedback.
> > >
> > > We are also glad to hear that our response resolves your concerns.
> > >
> > > We have incorporated additional clarifications and discussions on
> > > - Sigmoid shape
> > > - Comparisions with Wu et al., 2025 (W2)
> > > - Design and validation of false negative loss
> > >
> > > in our updated manuscripts and we believe these changes have significantly improved the clarity of our manuscript!
> > >
> > > Thanks,
> > >
> > > Authors

---

### Official Review · Reviewer_P5yB · 2026-03-13

**Soundness:** 3
**Presentation:** 2
**Significance:** 3
**Originality:** 3
**Overall Recommendation:** 4
**Confidence:** 4

**Summary:**

This paper frames adaptive early stopping for reasoning LLMs as a risk-control problem, where users specify acceptable error tolerance while the system calibrates stopping rules to meet this tolerance and minimize compute. The paper introduces a dual-threshold policy with an upper threshold that stops when the model suggests the questionis unlikely to be solved within the budget, plus a validation-based selection of signals and thresholds with finite-sample risk control and an efficiency-based tie-breaker. Experiments are conducted on four models and three datasets.

**Compliance With Llm Reviewing Policy:**

Affirmed.

**Final Justification:**

I maintain my positive score for this paper and highly recommend that the author include the rebuttal content in their final version.

**Key Questions For Authors:**

There’s a growing body of work that suggests shorter reasoning chains can be more robust than longer ones [1][2][3]. Does your early-stopping framework implicitly benefit from this? It might be worthwhile to discuss this in the final version.

[1] Hassid, et al. "Don't Overthink it. Preferring Shorter Thinking Chains for Improved LLM Reasoning." arXiv preprint arXiv:2505.17813 (2025).

[2] Xu, et al. "DTS: Enhancing Large Reasoning Models via Decoding Tree Sketching." arXiv preprint arXiv:2511.00640 (2025).

[3] Moshkov, et al. "Aimo-2 winning solution: Building state-of-the-art mathematical reasoning models with openmathreasoning dataset." arXiv preprint arXiv:2504.16891 (2025).

**Limitations:**

Please refer to the weakness

**Strengths And Weaknesses:**

Strengths:
- The experiments are relatively comprehensive, covering multiple datasets and including multimodal math. There is only one minor issue about the experiments that I will raise below.
- The paper considers adaptive thinking from a user-side risk-control perspective and formulates a principled framework, which is interesting in my opinion.

Weaknesses:

- I appreciate that the authors tested the model on four different models. However, all the models tested are based on Qwen (either the Qwen3 or the Deepseek distilled version). If the authors can provide an experiment on another model architecture, such as GPT-OSS or DeepSeek-R1-Distill-Llama on one or two datasets, it would strengthen the paper more.
- The lower threshold is defined as a sigmoid function of token usage. This appears somewhat ad-hoc. Can the authors justify why this form should reliably capture the lack of progress during reasoning?
- Although the paper is well written, the conclusion section is missing. The overall flow of the paper can be improved by adding it in the final version.

---

> ### Author Rebuttal · Authors · 2026-03-30
>
> We would like to thank the reviewer for the positive comments. We are glad the reviewer acknowledged the comprehensiveness of our experiments and found our novel risk control perspective interesting!
>
> # More models
>
> We’ve additionally included results on DeepSeek-R1-Distill-Llama on AIME! Here are the risk control validation results
>
> https://imgur.com/a/LZnu2Ec
>
> And the ensemble results:
>
> https://imgur.com/a/PopckOD
>
> # Sigmoid shape
>
> First, we would like to apologize that there is a typo in Eq.12. In our experiments, we actually have three parameters for the sigmoid:
>
> - The slope: Controls the steepness of the transition — a larger slope produces a sharper, more abrupt switch, while a smaller slope yields a more gradual curve.
> - The shift: Controls the location of the inflection point along the token axis, i.e., at which token count the transition occurs.
> - The lower bound: Sets the minimum value of the function, allowing the curve to start from a non-zero baseline.
>
> Y = sigmoid(x) * (1 - lower_bound) + lower_bound, x = (#tokens - shift) * slope
>
> It is only for the visualization part (Fig. 3), we keep the lower bound as 0, shift as B / 2 and only adjust the slope. For the actual early stopping experiments, all three parameters are calibrated on the validation set.
> These three parameters combined allow the resulting function to cover a very wide range of shapes beyond the sigmoid shape, for example:
> - A shift far to the left vs. right of the token budget places the inflection point outside the operating range, causing the function to behave as a purely convex vs. concave curve over the token range of interest.
> - A small slope, large shift, and positive lower bound cause the sigmoid to flatten out within the token range, effectively recovering a constant lower threshold.
>
> We apologize again for the typo and have corrected it in the revised manuscript.
>
> # Conclusion section
>
> Thanks! We have added a conclusion section summarizing the paper!
>
> # Questions
>
> Thank you for pointing out these works; they are definitely relevant!
> While our work is mainly targeting improving the efficiency through early stopping, i.e. same accuracy less tokens, it could certainly be possible that our framework can boost the accuracy via alleviating overthinking. We have included a discussion on this line of literature in an additional “future direction” section. In the paragraphs below, we provide a detailed summarization and comparison
>
> [1] Hassid et al. (2025). Their method is based on sampling multiple complete reasoning chains, stopping when the first m out of k chains finish, and then performing majority voting over those completed chains. Thus, their framework still relies on generating full trajectories for several independent samples and then aggregating them. By contrast, our method is instance-level early stopping within a single reasoning trajectory (or within an ensemble member) In particular, our method can directly reduce the length of an individual reasoning chain itself, rather than only preferring shorter completed samples among multiple candidates. So while [1] provides strong evidence that shorter completed chains can be preferable, our contribution is a different mechanism.
>
> [2] Xu et al. (2025, DTS). This paper is also related, but differs in mechanism and objective. DTS builds a decoding tree, selectively branches at high-entropy tokens, and then uses early stopping to identify a short completed reasoning path within that branched search procedure. In other words, DTS is a search/decoding algorithm designed to sketch the reasoning tree and find short successful trajectories. Our framework does not redesign decoding in this way, does not require tree expansion or branching, and does not aim to approximate the shortest correct path. Instead, we study the orthogonal problem of how to stop reasoning with statistical risk guarantees under a compute budget. Our stopping thresholds are calibrated on validation data to satisfy a user-specified error target, whereas DTS is primarily motivated by improving search efficiency and accuracy through structured exploration of the reasoning space. So [2] is highly relevant as further evidence that overthinking is real, but it is still quite different from our setting.
>
> [3] Moshkov et al. (2025). We agree this paper is related mainly through its empirical observation that shorter reasoning chains often tend to be more accurate. We view this as an important motivating observation for our work. At the same time, [3] does not propose the same kind of stopping framework as ours; rather, it is part of a broader reasoning-system pipeline involving data construction, tool-integrated reasoning, and candidate selection. We will therefore cite it in the final paper as supporting motivation for why adaptive stopping is meaningful, and clarify that our contribution is to turn this intuition into a principled stopping rule with explicit risk control.

---

> > ### Author Rebuttal · Reviewer_P5yB · 2026-04-01
> >
> > Thank you for your detailed responses to my questions. Given the current situation, the updated information on the formulation correction, and the paper's writing quality, I still lean toward maintaining my current positive score for acceptance. I would also like to strongly encourage the authors to include all the content from the rebuttal in the updated version, which will make the paper easier to understand, especially regarding the sigmoid shape part.

---

> > > ### Author Response · Authors · 2026-04-01
> > >
> > > We would like to thank the reviewer again for the careful review and constructive feedbacks!
> > >
> > > We agree that the discussion in the rebuttal phase has greatly improved the clarity and quality of the paper, and we will certainly include it in our manuscript!
> > >
> > > Thanks,
> > >
> > > Authors

---

### Official Review · Reviewer_mCrS · 2026-03-13

**Soundness:** 2
**Presentation:** 3
**Significance:** 3
**Originality:** 3
**Overall Recommendation:** 5
**Confidence:** 3

**Summary:**

The paper frames early stopping for reasoning LLMs as risk control, introducing dual upper and lower thresholds calibrated on a validation set with finite-sample correction. The upper threshold stops when the model is confident, and the lower threshold stops when the model is not making progress. Empirical results show the framework reliably meets user-specified risk tolerances and improves token efficiency.

**Compliance With Llm Reviewing Policy:**

Affirmed.

**Final Justification:**

The rebuttal addressed several of my concerns and questions. The mismatch between Eq12 in original submission and the actual parameter implementation used in experiments should be fully corrected in the final manuscript. I keep my positive impression of the paper and recommend acceptance.

**Key Questions For Authors:**

How sensitive are the results to the choice of maximum budget B and chunk granularity, and does the sigmoid shape need to be re-tuned when these change?

Is there evidence sigmoid lower threshold shape is appropriate across tasks and models, and what happens when confidence dynamics peak early or late in the trajectory?

**Limitations:**

Yes

**Strengths And Weaknesses:**

**Strength**

The paper identifies the gap that prior work only handles confident success, and the paper correctly points out that on hard datasets most instances never reach the upper threshold, wasting the entire budget. The dual-threshold framing directly addresses this. Recasting threshold tuning as specifying an interpretable error tolerance is a clean and useful.

The finite-sample correction via UCB is backed by Hoeffding's inequality and Figure 4 shows it prevents the naive approach from violating the target risk, especially under small validation sets.

The ensemble selection picks the most efficient signal per epsilon rather than committing to one upfront, and Figure 5 shows this transfers to the test set.

**Weaknesses**

The lower threshold guarantee breaks when combined with the upper threshold. After the lower threshold abstrains instances, the false positive rate over remaining instances can increase or decrease unpredictably and the paper does not quantify how often or how badly this happens in practice.

Solvability labels in Figure 6 are defined by whether the model reaches the correct answer under the full budget, making them depend on each other. The threshold is calibrated using a notion of solvability that itself depends on the budget you are trying to reduce. This makes the efficiency gains in Figure 6 hard to interpret.

The sigmoid lower threshold has a rigid form centered at the budget midpoint with a single slope parameter, with no justification or ablation for why this shape is appropriate across tasks and models where confidence dynamics can peak early or late.

The ensemble result in Figure 5 is dominated by the probe model on Qwen3-8B, but the probe requires labeled AIME trajectories to train and no probe is available for other datasets, which limits how general the ensemble contribution actually is.

---

> ### Author Rebuttal · Authors · 2026-03-30
>
> We thank the reviewer for the thorough and technically sharp reading. We are glad the reviewer recognized the dual-threshold framing as directly addressing a real gap, the UCB correction as principled, and the ensemble selection as transferable. We address each weakness below.
> # Weakness
>
> ## Break of false positive loss / upper threshold loss
>
> >The lower threshold guarantee breaks when combined with the upper threshold
>
> To clarify: it is the upper threshold (false positive rate) guarantee that can break, not the lower threshold.
>
> The false positive rate is defined as:
> $$\text{FPR} = \frac{I}{I + C}$$
> where $I$ and $C$ denote the number of incorrectly and correctly answered queries, respectively.
>
> If the lower threshold causes us to abstain from fractions $p_i$ and $p_c$ of incorrect and correct samples, respectively, the new FPR becomes:
>
> $$\text{FPR}_{\rm new} = \frac{(1 − p_i)I}{(1 − p_i)I + (1 − p_c)C}$$
>
> It follows that when $p_i < p_c$ — i.e., the lower threshold filters out proportionally more correct samples than incorrect ones — we have FPR_new > FPR, meaning the guarantee breaks. Conversely, when $p_i \geq p_c$, the guarantee holds.
>
> This failure mode arises when the lower threshold mechanism itself fails, for example, due to a significant distribution shift between the calibration and test sets. In such cases, the lower threshold may have been well-calibrated but generalizes poorly, causing it to disproportionately abstain on correct samples.
> It is worth noting, however, that distribution shift tends to break all guarantees, not just this one. We have added an explicit limitations section discussing this (see also our response to reviewer RnRS).
>
> In practice, as long as the lower threshold functions as intended — filtering out more incorrect samples than correct ones ($p_i \geq p_c$) — the upper threshold guarantee will remain intact. The failure mode described above is therefore an edge case tied to severe distribution shift, rather than a concern under normal operating conditions. To confirm this, we empirically evaluated the guarantee's robustness on a real dataset (Qwen3-8B, AIME + GPQA 1:1, consistent with the Fig. 6 setting), fixing the upper threshold while varying the lower threshold ε. The results show that the guarantee holds reliably across a wide range of ε values:
> https://imgur.com/a/2NeDxky
>
>
> ## Solvability label
>
> The message behind Fig 6 is to demonstrate how lower and upper thresholds help improve efficiency in datasets of different difficulties. And since most datasets do not have a difficulty label provided, we have to define the dataset difficulty as the portion of samples that can be solved in a fixed (and large) reasoning budget (the solvability label).
>
> ## Ensemble
>
> We only trained probe model for Qwen3-8B (top left grid of Fig.5).
>
> We agree that probing is expensive as it is a supervised approach. And since it saves the most tokens, the ensemble would pick it when available.
> In the rest grids of Fig.5, we did not train a probe model, ensemble picked some other signals.
>
> # Questions
>
> ## Sensitivity to chunk and maximum budget
>
> For chunk granularity, no, as the x-axis of the lower threshold function is #tokens rather than chunk index, so adjusting
>
> When the maximum budget is changed, we do expect the sigmoid shape to be re-tuned.
> For example, when the budget is increased, there could be more questions solved and so we need to abstain less samples to maintain the same false negative rate therefore the slope needs to be decreased.
>
>
> ## Sigmoid shape
>
> First, we would like to apologize that there is a typo in Eq.12. In our experiments, we actually have three parameters for the sigmoid:
>
> - The slope: Controls the steepness of the transition, a larger slope produces a sharper, more abrupt switch, while a smaller slope yields a more gradual curve.
> - The shift: Controls the location of the inflection point along the token axis, i.e., at which token count the transition occurs, which can adapt to *confidence dynamics peaking early or late*.
> - The lower bound: Sets the minimum value of the function, allowing the curve to start from a non-zero baseline.
>
> Y = sigmoid(x) * (1 - lower_bound) + lower_bound, x = (#tokens - shift) * slope
>
> It is only for the visualization part (Fig. 3), we keep the lower bound as 0, shift as B / 2 and only adjust the slope.
> For the early stopping experiments, however, we search over all three parameters. This added flexibility allows the resulting function to cover a much wider range of shapes, for example:
> - A shift far to the left vs. right of the token budget places the inflection point outside the operating range, causing the function to behave as a purely convex vs. concave curve over the token range of interest.
> - A small slope, large shift, and positive lower bound cause the sigmoid to flatten out within the token range, effectively recovering a constant lower threshold.
>
> We apologize again for the typo and have corrected it in the revised manuscript.

---

> > ### Author Rebuttal · Reviewer_mCrS · 2026-04-01
> >
> > Thank you to the authors for rebuttal and explanations.
> >
> > Since experimentals have 3 parameters for the sigmoid, I think the paper should discuss these parameters and their implementation alongside the corrected Eq12. Also, given this fix, Figure 3 is then a simplification, as 2 parameters are fixed. That figure also could benefit from discussing the typical selected parameter values and confirming that the shapes in Fig. 3 are representative of what Algorithm 1 selects in practice.
> >
> > My other concerns are largely resolved. I will maintain my original score, but will remain open during the reviewer AC discussion

---

> > > ### Author Response · Authors · 2026-04-07
> > >
> > > We would like to thank the reviewer for the detailed response.
> > >
> > > We have already revised our manuscript, correcting the sigmoid formula.
> > >
> > > > That figure also could benefit from discussing the typical selected parameter values and confirming that the shapes in Fig. 3 are representative of what Algorithm 1 selects in practice.
> > >
> > > Thanks for the feedback! We have included additional discussion in the caption as well as below the sigmoid equations.
> > >
> > > We have also included an extra figure in the appendix, visualizing the parametric functions acquired from Algorithm 1 under different false negative tolerance levels, under the same setting as Figure 6: https://imgur.com/a/jxaE4g3
> > >
> > > One can see that under the three parameters, sigmoid parameterization, the lower threshold shows very diverse shapes under different values of $\epsilon$

---

### Official Review · Reviewer_RnRS · 2026-03-15

**Soundness:** 3
**Presentation:** 4
**Significance:** 3
**Originality:** 4
**Overall Recommendation:** 4
**Confidence:** 3

**Summary:**

The paper introduces a framework for controlling the compute used by reasoning LLMs by turning early stopping into a risk-controlled decision problem. Specifically, it allows the user to specify an acceptable error rate, and the system calibrates stopping rules using a validation set so that the realised risk remains below this target. The method is supported by a dual-threshold mechanism: an upper confidence threshold that stops reasoning once the model appears confident in its answer, and a lower threshold that stops reasoning when progress is unlikely. Experiments across reasoning models and datasets show that this approach can significantly reduce token usage while maintaining similar accuracy.

**Compliance With Llm Reviewing Policy:**

Affirmed.

**Final Justification:**

I acknowledge the rebuttal by the authors. The rebuttal has addressed most of my concerns. Hence, I stand by my positive impression of the paper and recommend weak accept.

**Key Questions For Authors:**

1.  How sensitive is the framework to the quality of the uncertainty signals, and does performance degrade significantly when these signals are weakly correlated with correctness? Can you show an experiment where this is the case or if that's not possible, explain to me what you'd expect to happen?
1. How large must the validation set be for reliable calibration, and how frequently would recalibration be required in realistic deployments ? Does this calibration (even partially) carry from one domain to another? Importantly, what about a very sparse setting/domain?
1. How well do calibrated thresholds transfer when the deployment distribution differs substantially from the calibration distribution? (Maybe this question is answered in through the previous ones as well, but I think it's an important one so I'd like it as a standalone as well).

**Limitations:**

The authors do not discuss the limitations. I believe the should. Very briefly:
- Reliance on uncertainty signals
- Distribution shift risks
- Deployment trade-offs

**Strengths And Weaknesses:**

# Strengths
1. The paper reframes early stopping in reasoning LLMs as a risk-control problem, an approach allowing users to specify an error tolerance instead of tuning thresholds or token budgets. This approach makes sense, it's original and I assume to be more interpretable for some users.
1. The idea of expressing compute decisions in terms of risk tolerance offers a useful perspective that may influence future work on adaptive inference.

# Weaknesses
1. Line 364: The experiment setup is limited when it comes to the models and dataset aspects. In terms of models, the authors have only tested their approach on Qwen models or DeepSeek-R1 distilled to Qwen. I would like to see how the approach generalises across different models. In terms of datasets, the paper mainly focuses on math datasets. I would like to see how the approach generalises across different tasks such as code and planning. My next point, gives further explanation as to why I believe the method needs to be further tested.
1. The framework relies on uncertainty or probe-based signals, and its effectiveness may vary depending on how well these signals correlate with correctness or reasoning progress. Hence, experimenting across different domains where such signals vary is very important.
1. Risk guarantees require the framework to be calibrated.
1. The calibration procedure and signal extraction may introduce additional costs or engineering complexity that are not extensively discussed.
1. No code.
1. Minor - Line 377: Good to cite vLLM since you're referring to it. Same for the AIME dataset --> Especially since the code is not provided, it's important to know exactly what dataset was used.

---

> ### Author Rebuttal · Authors · 2026-03-30
>
> We thank the reviewer for the careful reading and constructive feedback. We are encouraged that the reviewer found our reframing of early stopping as a risk-control problem to be "original" and "more interpretable", and noted that "the idea of expressing compute decisions in terms of risk tolerance offers a useful perspective that may influence future work on adaptive inference." We are also pleased the reviewer rated Presentation as *Excellent (4)* and Originality as *Excellent (4)*. We address each remaining concern below.
>
> We now address your comments:
>
> # Weakness
>
> ## W1, W2 and Q1
>
> This is a very good question. There will certainly be problems where the uncertainty signal fails, e.g. in the EAT paper [1], the authors admit that their signal may not be effective for long-horizon generation tasks such as coding.
>
> However, this does not pose challenges to our framework: Consider a signal uncorrelated with the correctness, our Algorithm 1 will simply choose to return a loose and unachievable threshold value, as any early stopping condition based on this signal would fail to meet the user’s target risk level.
>
> [1] https://arxiv.org/abs/2509.26522
>
>
> On signal quality (Q1): there will certainly be problems where uncertainty signals fail — the EAT paper [1] explicitly acknowledges their signal may not be effective for long-horizon tasks such as coding. However, this does not pose challenges to our framework: for a signal *uncorrelated with correctness,* Algorithm 1 will simply return a loose threshold, as any early stopping condition based on this signal would fail to meet the user’s target risk level — naturally falling back to the token-budget baseline. Guarantees remain valid.
>
> ## W3: Risk guarantees require calibration
>
> We respectfully note that this is *by design*: the calibration procedure *is* how we provide guarantees. Crucially, we provide a principled, statistically grounded calibration (Algorithm 1) that requires only a lightweight forward pass over a small validation set — no gradient computation — taking minutes in practice. This is precisely the advantage over prior work, which relies on parameter tuning with no guarantees.
>
>
> ## W4
>
> We agree that signal extraction and the construction of a calibration set can introduce extra implementation and storage complexity in the pipeline. We have added an additional section discussing the complexity.
>
> But the calibration procedure (Alg.1) is very lightweight and efficient, where only NumPy coding is involved.
>
> ## W5
>
> We have uploaded the code here: https://anonymous.4open.science/r/reasoning_risk_control-FDB9/
>
> ## W6
>
> Thanks! We have added the citation.
>
>
> ## Q2 Size of the validation set
>
> In Fig. 7, we already provided an ablation study on the validation set size: the results show that, a validation set size larger than 16 seems to be sufficient to control the risk. Space permitting, we will move this to the main text in the revision.
>
> Note that with the risk control framework, a small validation set in practice would not compromise the risk guarantee as the finite sample correction (Eq 15, 23) would make the threshold more conservative when the validation set is small.
>
> We also note that calibration is a one-time offline procedure requiring only a forward pass — no gradient computation — taking minutes in practice.
>
> ## Q2 and Q3: Distribution shift and recalibration
>
> We ablate distribution shift in Fig. 8 and 9 in the appendix.
>
> We believe the transferability of thresholds under distribution shift depends on the exact types of distribution shift:
> For *topic* shift, the threshold picked from our risk control, framework seems fairly robust, much more robust than the naive cross validation. We hypothesize that this is because the uncertainty signal’s statistics, e.g. magnitude / variance, behaves similarly across datasets.
> For *length* shift, in particular, when the validation set’s reasoning length is shorter than the test set (Fig.8 left), our framework shows less robustness, especially on the lower threshold’s false negative loss, whose shape is dependent on the reasoning horizon.
>
> We acknowledge that the risk control framework itself provides no formal guarantee under distribution shift, and will include this as an explicit limitation.
>
> # Limitations
>
> Thanks! We have included an extra section on the limitations.

---

> > ### Author Rebuttal · Reviewer_RnRS · 2026-04-03
> >
> > I thank the authors for their detailed rebuttal.
> >
> > Almost all of my concerns have been addressed. The remaining points remain though: How does the method generalize (a) when different models are used and (b) across different type of tasks.
> >
> > I appreciate the authors' response, but I am actually more concerned after reading it. The authors confirm that when uncertainty signals are uninformative, the method simply falls back to the token-budget baseline. While I understand this preserves theoretical guarantees, it also means the method provides zero practical benefit in precisely the domains I was asking about (coding, planning, long-horizon reasoning). The rebuttal frames graceful degradation as a strength, but to me it highlights a significant limitation in the method's generalizability. My original concern was not about whether guarantees break, but about whether the method is practically useful beyond math. The response suggests it may not be, and the lack of any empirical evidence on non-math domains reinforces this. Finally, the rebuttal does not address the model diversity concern: all experiments use Qwen models or DeepSeek-R1 distilled to Qwen.
> >
> > I will therefore maintain my score.

---

> > > ### Author Response · Authors · 2026-04-03
> > >
> > > We would like to thank the reviewer for the additional feedback and comments. We agree that only having Qwen + Math lacks diversity for the empirical verification.
> > >
> > > Therefore, we have included additional experiments testing our framework on extra models and applications **beyond just Qwen + Math**:
> > >
> > > # Other models
> > >
> > > We performed an additional evaluation on two *non-Qwen* models, where the risk control guarantee (first links) and the ensemble of signals remain effective (second links):
> > >
> > > - DeepSeek distilled Llama: https://imgur.com/a/LZnu2Ec  https://imgur.com/a/PopckOD
> > >
> > > - Phi-4-reasoning: https://imgur.com/a/gERjI78 https://imgur.com/a/xFlj2kk
> > >
> > > where Phi-4 is a reasoning model unaffiliated with Qwen or DeepSeek in any form.
> > >
> > > # Other tasks
> > >
> > > We have additionally included two *non-math* datasets using EAT [1] as the uncertainty signal, where the risk control guarantee still holds
> > >
> > > - DeepSeek-R1-0528-Qwen3-8B on 60 questions from MMLU astronomy: https://imgur.com/a/NgSVsOB
> > >
> > > - Qwen3-4B-Thinking-2507 Tool calling: https://imgur.com/a/kGC6WXs
> > >
> > > [1] https://arxiv.org/abs/2509.26522
> > >
> > > # Coding, planning, long-horizon reasoning
> > >
> > > Firstly, we want to emphasize that risk control is fundamentally a procedure that jointly calibrates an uncertainty signal and a user-provided risk target such that they are compatible.  If either of those inputs is bad, then early stopping won’t be feasible. It is also worth noting that we believe developing a new uncertainty signal for a particular task goes beyond the scope of the current submission.
> > >
> > > Regarding these three specific domains, we are currently unaware of informative uncertainty signals applicable to them. As a result, our framework cannot yet enable principled early stopping in these settings, since it is fundamentally predicated on the availability of such signals. We acknowledge this is a meaningful limitation, and we will state it explicitly in the revised manuscript.
> > >
> > > That said, we are optimistic about the trajectory of this area. Uncertainty-based early stopping is still a growing area, most of the signals we build upon were released only after June 2025, and we anticipate that future work will develop suitable signals for these harder reasoning domains. Our framework is designed to be modular with respect to the choice of signal, so we expect it can be readily extended to these settings once informative signals become available.
> > >
> > > We want to be careful not to overclaim, however, and so rather than framing this as a forthcoming benefit, we prefer to treat the absence of applicable signals for coding, planning, and long-horizon reasoning as an explicit limitation of the current work.

---

### Decision · Program_Chairs · 2026-04-30

**Decision:**

Accept (regular)

**Comment:**

This paper studies LLM reasoning through the lens of uncertainty quantification. Based on a user-specified allowable error rate, the method calibrates a reasoning stopping rule so that the realized risk remains below the user's target rate. The method is based on an upper confidence threshold that stops reasoning when the model is confident about its answer, together with a lower threshold that stops reasoning when further progress is unlikely. The authors engaged substantively during the author response period and almost all reviewer concerns were resolved. The main lingering concern was from Reviewer RnRS about real-world applicability to coding, planning, and long-horizon reasoning. The authors said they will explicitly narrow the scope in the revision, and they should make all other promised revisions as well in the camera-ready.